# Suppression of unwanted CRISPR-Cas9 editing by co-administration of catalytically inactivating truncated guide RNAs

John C. Rose [1,9,12 ✉], Nicholas A. Popp [2,12], Christopher D. Richardson [3,4,10], Jason J. Stephany[2], Julie Mathieu[5], Cindy T. Wei [1], Jacob E. Corn [3,4,11], Dustin J. Maly [1,6 ✉] & Douglas M. Fowler [2,7,8 ✉]

CRISPR-Cas9 nucleases are powerful genome engineering tools, but unwanted cleavage at off-target and previously edited sites remains a major concern. Numerous strategies to reduce unwanted cleavage have been devised, but all are imperfect. Here, we report that off-target sites can be shielded from the active Cas9•single guide RNA (sgRNA) complex through the co-administration of dead-RNAs (dRNAs), truncated guide RNAs that direct Cas9 binding but not cleavage. dRNAs can effectively suppress a wide-range of off-targets with minimal optimization while preserving on-target editing, and they can be multiplexed to suppress several off-targets simultaneously. dRNAs can be combined with high-specificity Cas9 variants, which often do not eliminate all unwanted editing. Moreover, dRNAs can prevent cleavage of homology-directed repair (HDR)-corrected sites, facilitating scarless editing by eliminating the need for blocking mutations. Thus, we enable precise genome editing by establishing a flexible approach for suppressing unwanted editing of both off-targets and HDR-corrected sites.

[1] Department of Chemistry, University of Washington, Seattle, WA 98195, USA. [2] Department of Genome Sciences, University of Washington, Seattle, WA 98195, USA. [3] Innovative Genomics Initiative, University of California, Berkeley, Berkeley, CA 94720, USA. [4] Department of Molecular and Cell Biology, University of California, Berkeley, Berkeley, CA 94720, USA. [5] Department of Comparative Medicine, Institute for Stem Cell and Regenerative Medicine, University of Washington, Seattle, WA 98109, USA. [6] Department of Biochemistry, University of Washington, Seattle, WA 98195, USA. [7] Department of Bioengineering, University of Washington, Seattle, WA 98195, USA. [8] Genetic Networks Program, Canadian Institute for Advanced Research, Toronto, ON, Canada. [9] Present address: Center for Personal Dynamic Regulomes, Stanford University School of Medicine, Stanford, CA 94305, USA. [10] Present address: Department of Molecular, Cellular, and Developmental Biology, University of California, Santa Barbara, CA 93106, USA. [11] Present address: Institute of Molecular Health Sciences, Department of Biology, ETH Zurich, Zurich, Switzerland. [12] These authors contributed equally: John C. Rose, Nicholas A. Popp. ✉email: jackrose@stanford.edu; djmaly@uw.edu; dfowler@uw.edu

The *S. pyogenes* Cas9 (SpCas9) nuclease is targeted to specific sites in the genome by a single guide RNA (sgRNA) containing a 20-nucleotide target recognition sequence. The target site must also contain an NGG protospacer adjacent motif (PAM)[1]. This multipartite target recognition system is imperfect, and most sgRNAs direct significant cleavage and subsequent unwanted editing at off-target sites whose sequence is similar to the target site[2–5]. Numerous approaches to reduce off-target editing have been devised, yet are hampered by various limitations[6–17]. For example, SpCas9 variants with improved specificity have been engineered[18–20]. While useful, these high-specificity variants often decrease on-target editing[21,22] and in most cases do not eliminate all unwanted editing[20]. All high-specificity Cas9 variants appear to balance on- vs off-target activity via the same mechanism[20,23] and, as a consequence, often fail to suppress editing at the same obstinate off-target sites[20,22]. Thus, new methods for off-target suppression are needed, particularly ones that preserve on-target editing, can be combined with high-specificity Cas9 variants, and require minimal expenditure of time, effort, and resources. To this end, we developed an orthogonal and general approach for suppressing off-targets that can be readily combined with existing methods, including high-specificity variants.

Our off-target suppression approach is based on the observation that sgRNAs with target recognition sequences 16 or fewer bases in length direct Cas9 binding to DNA target sites but do not promote cleavage[24–26]. Here, we show that Cas9 bound to dRNAs with perfect complementarity to off-target sites can dramatically improve editing specificity by shielding these sites from the active Cas9•sgRNA complex (Fig. 1a). To highlight the generality and ease of implementation of our method, which we call dRNA Off-Target Suppression (dOTS), we effectively suppress editing at 15 off-target sites, yielding up to a ~40-fold increase in specificity, with minimal dRNA optimization. Furthermore, dOTS can be multiplexed to suppress several off-targets simultaneously and can be combined with other approaches for improving specificity. We also describe dRNA ReCutting Suppression (dReCS), wherein dRNAs prevent recutting of homology-directed repair (HDR)-corrected sites, eliminating the need for blocking mutations and facilitating scarless editing. Thus, we enable more precise genome editing by establishing a facile and flexible approach for suppressing unwanted editing of both off-target and HDR-corrected sites.

## Results

**Dead RNA off-target suppression increases specificity**. We first determined the feasibility of using dRNAs to suppress unwanted editing at off-target site 1 (OT1) of an sgRNA (sgRNA2) targeting the *FANCF* locus[18]. We co-transfected HEK-293T cells with a plasmid encoding SpCas9, along with equal amounts of plasmids encoding *FANCF*-sgRNA2 and a GFP control, or *FANCF*-sgRNA2 and one of four dRNAs with perfect complementarity to OT1 (Supplementary Fig. 1a). Three of the four dRNAs significantly decreased off-target editing without appreciably impacting on-target editing, while co-transfection of a nontargeting control dRNA did not impact on- or off-target editing (Supplementary Fig. 1b). In particular, dRNA1 decreased off-target editing from 20.44% (s.e.m. = 0.61%, $n = 3$) to 0.69% (s.e.m. = 0.02%, $n = 3$), leading to a 30-fold increase in the on-target specificity ratio (Fig. 1b). Cas9•dRNA complexes are thought to lack cleavage activity, but a relatively small number of dRNAs have been evaluated so far[24,25]. Thus, we verified that dRNA1 did not direct any detectable Cas9 editing activity at either the on- or off-target sites (Supplementary Fig. 1c). We further confirmed that dRNA1 showed no cleavage genome-wide using GUIDE-seq[5], and that it directed selective reduction of only

OT1 (Supplementary Fig. 2). To our knowledge, this experiment is the first to demonstrate that a dRNA leads to no detectable cleavage activity anywhere in the genome.

To demonstrate the generality of dOTS, we evaluated 18 additional on-target/off-target pairs in HEK-293T cells. We found at least one dRNA for 15 of the 19 pairs we tested that increased the specificity ratio by at least two-fold (mean fold-change = 10.44) while decreasing on-target editing by no more than

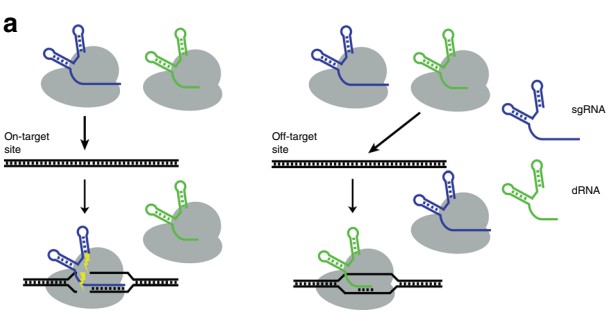

**a**

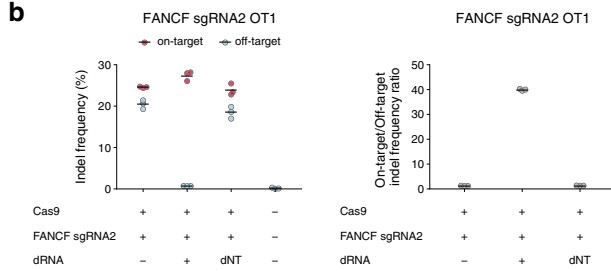

**b**

FANCF sgRNA2 OT1

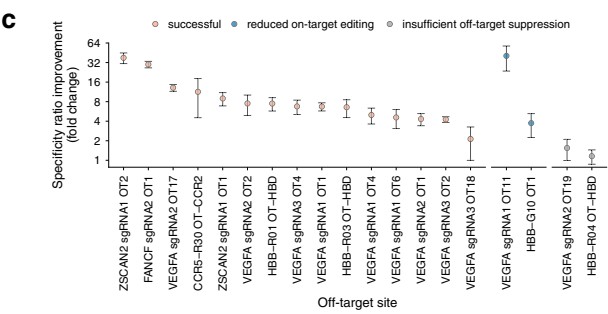

**c**

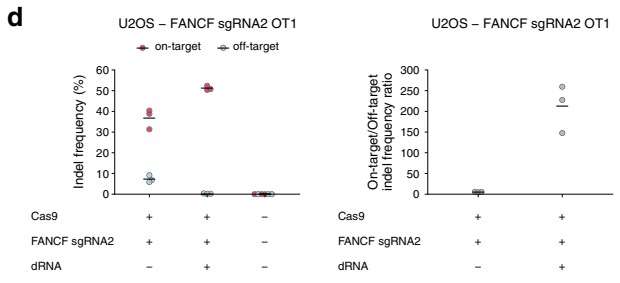

**d**

U2OS – FANCF sgRNA2 OT1

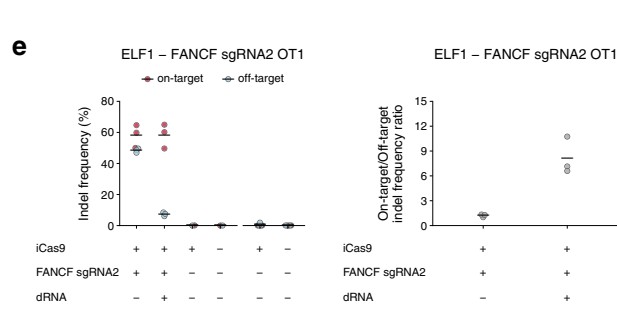

**e**

ELF1 – FANCF sgRNA2 OT1

**Fig. 1 dRNA-mediated Off-Target Suppression (dOTS) effectively reduces off-target editing. a** Schematic representation of dOTS. A dRNA (green) with perfect complementarity for an off-target site directs Cas9 binding but not cleavage, protecting the site. **b** Indel frequencies and specificity ratios (on-target/off-target indel frequency ratios) at the *FANCF*-sgRNA2 on-target site and OT1 24 h after transfection of HEK-293T cells with Cas9, sgRNA, and *FANCF*-sgRNA2 OT1 dRNA1 or a nontargeting control dRNA (dNT) that does not target genomic DNA. For conditions without dRNA, an equivalent amount of pMAX-GFP was substituted. Means of $n = 3$ biological replicates depicted by solid lines. **c** Normalized specificity ratios, computed as the specificity ratio of the best dRNA condition (Supplementary Table 1) divided by the specificity ratio of the sgRNA only condition for 19 guide/off-target pairs tested in HEK-293T cells. Points depict the mean of $n = 3$ biological replicates, error bars show the standard error of the mean. **d** Indel frequencies and specificity ratios at the *FANCF*-sgRNA2 on-target site and OT1 24 h after transfection in U2OS cells, and **e** Elf1 embryonic stem cells. Control samples to the right of the x-axis break were performed separately. iCas9 denotes stable integration of Cas9 under the control of a doxycycline-inducible promoter. Means of $n = 3$ cell culture replicates depicted by solid lines. OT off-target. Source data are available in the Source Data file.

two-fold (mean fold-change = 0.93; Fig. 1c; Supplementary Fig. 3). Across all on-target/off-target pairs, a median of six candidate dRNAs were screened, highlighting the ease of identifying effective dRNAs (Supplementary Figs. 3 and 4; Supplementary Table 1). In most cases, nontargeting dRNAs had little to no impact on editing (Supplementary Fig. 5). Moreover, effective dRNAs did not induce indels at either on- or off-target sites, suggesting that few, if any, Cas9•dRNA complexes are active (Supplementary Tables 2 and 3). dOTS was also effective in U2OS cells and the Elf1 naïve embryonic stem cell line (Fig. 1d, e; Supplementary Fig. 6)[27]. Finally, we found that dRNA-mediated suppression of off-target editing was durable, with dRNAs effectively decreasing off-target editing for at least 72 h post-transfection (Supplementary Fig. 7).

An important application of Cas9 is editing genes containing pathogenic mutations[28,29]. For example, Cas9 has been used to target the $\beta$-globin locus (*HBB*), with the goal of curing sickle cell disease[30,31]. However, the $\delta$-globin locus (*HBD*) is a common off-target for sgRNAs targeting *HBB*, and cleavage of both on- and off-target sites can result in large chromosomal deletions at the globin locus[32]. In HEK-293T cells, dOTS decreased off-target editing at *HBD* from 1.08% (s.e.m. = 0.22%, $n = 3$) to 0.15% (s.e.m. = 0.03, $n = 3$; Supplementary Fig. 3d). In Elf1 cells, dOTS decreased off-target editing at *HBD* from 20.72% (s.e.m. = 2.75, $n = 3$) to 1.20% (s.e.m. = 0.18, $n = 3$), increasing the specificity ratio from 1.33 to 13.72 (Supplementary Fig. 6b). Thus, dOTS can control unwanted editing at clinically relevant loci.

We were unable to find effective dRNAs for four off-target sites. In two cases, dRNAs strongly reduced off-target editing but also decreased on-target editing by greater than two-fold (Fig. 1c; Supplementary Fig. 3b, i). In two other cases, no dRNA we tested was effective in decreasing off-target editing (Fig. 1c; Supplementary Fig. 3e, m, n). We suspect that these ineffective dRNAs are either unstable, form unfavorable secondary structures, or have insufficient affinity for the off-target site relative to their cognate sgRNAs. However, at most off-targets we identified one or more effective dRNAs that enhanced specificity without sacrificing on-target editing, making dOTS an effective approach for off-target suppression.

**Mechanism of off-target suppression by dRNAs.** dOTS is based on our prediction that Cas9•dRNA complexes with perfect

complementarity to an off-target site can directly outcompete active, imperfectly complementary Cas9•sgRNA complexes for binding. To test this Cas9 self-competition mechanism, we performed in vitro cleavage assays with linear DNA substrates and purified Cas9 ribonucleoprotein complexes (RNPs) containing either *FANCF*-sgRNA2 or dRNA1. Incubation of a substrate containing the *FANCF* OT1 site with a mixture of the Cas9•dRNA1 and Cas9•sgRNA2 complexes led to a robust reduction in cleavage compared to administration of the Cas9•sgRNA2 complex alone (Fig. 2a). Consistent with our self-competition mechanism, preincubation of the substrate with the Cas9•sgRNA2 complex followed by addition of the Cas9•dRNA1 complex eliminated the reduction in cleavage (Supplementary Fig. 8). Thus, Cas9•dRNA complexes can directly shield off-target loci from Cas9•sgRNA cleavage.

At low concentrations of Cas9•sgRNA2, Cas9•dRNA1 modestly reduced cleavage of the on-target *FANCF* substrate site in vitro (Fig. 2b), despite this dRNA not affecting on-target editing efficiency in cells (Fig. 1b, d, e). One possible explanation for this disparity is that, in cells, Cas9•dRNA1-mediated protection of the on-target locus decreases the rate of indel formation but editing reaches the same maximum as in cells without dRNA1 by the time of measurement. Another explanation is that cellular factors prevent Cas9•dRNA1, which should have modest affinity for the on-target site, from providing appreciable protection from cleavage by Cas9•sgRNA2. Thus, we measured rates of indel formation at *FANCF*-sgRNA2 OT1 and the on-target site in cells using a chemically inducible Cas9 (ciCas9) variant[6,33]. The activity of ciCas9 is repressed by an intramolecular autoinhibitory switch. Addition of a small molecule, A-1155463 (A115), disrupts autoinhibition and rapidly activates ciCas9, enabling precise studies of editing kinetics.

As expected, activation of ciCas9 with A115 led to the rapid appearance of indels at the *FANCF*-sgRNA2 on- and off-target sites in the absence of dRNA1. Inclusion of a plasmid encoding dRNA1 effectively eliminated ciCas9-mediated editing at the off-target site but had no measurable impact on the kinetics of on-target editing (Fig. 3a). These results suggest that dRNAs with imperfect complementarity to an on-target site can bind to and protect that site in cell-free systems, but not in cells. The most likely explanation for this difference is that, in cells, DNA is subject to a variety of active processes that influence Cas9[34,35]. For example, the degree of complementarity between a guide and its target affects the ability of polymerases to displace dCas9 from DNA[36], suggesting that polymerases may limit the ability of imperfectly complementary Cas9•dRNA complexes to shield on-target sites.

Our proposed Cas9 self-competition mechanism predicts that the level of off-target shielding provided by moderately effective dRNAs can be improved by manipulating the ratio of Cas9•dRNA to Cas9•sgRNA in cells. While an initial 1:1 plasmid ratio was effective for all 15 successful dRNAs, increasing the amount of dRNA relative to sgRNA further decreased off-target editing and improved the specificity ratio at each of the four sgRNA/dRNA pairs we tested (Fig. 3b, Supplementary Fig. 9). For one pair, higher dRNA:sgRNA ratios also decreased on-target editing. Thus, a trade-off between maintaining on-target editing and decreasing off-target editing exists for some sgRNA/dRNA pairs. Here, the dRNA/sgRNA ratio can be tuned based on whether preserving on-target editing or suppression of a particular off-target is desired.

**dOTS improves other approaches to increase Cas9 specificity.** Other strategies to improve Cas9 specificity fail to completely suppress off-target editing and often reduce on-target efficacy.

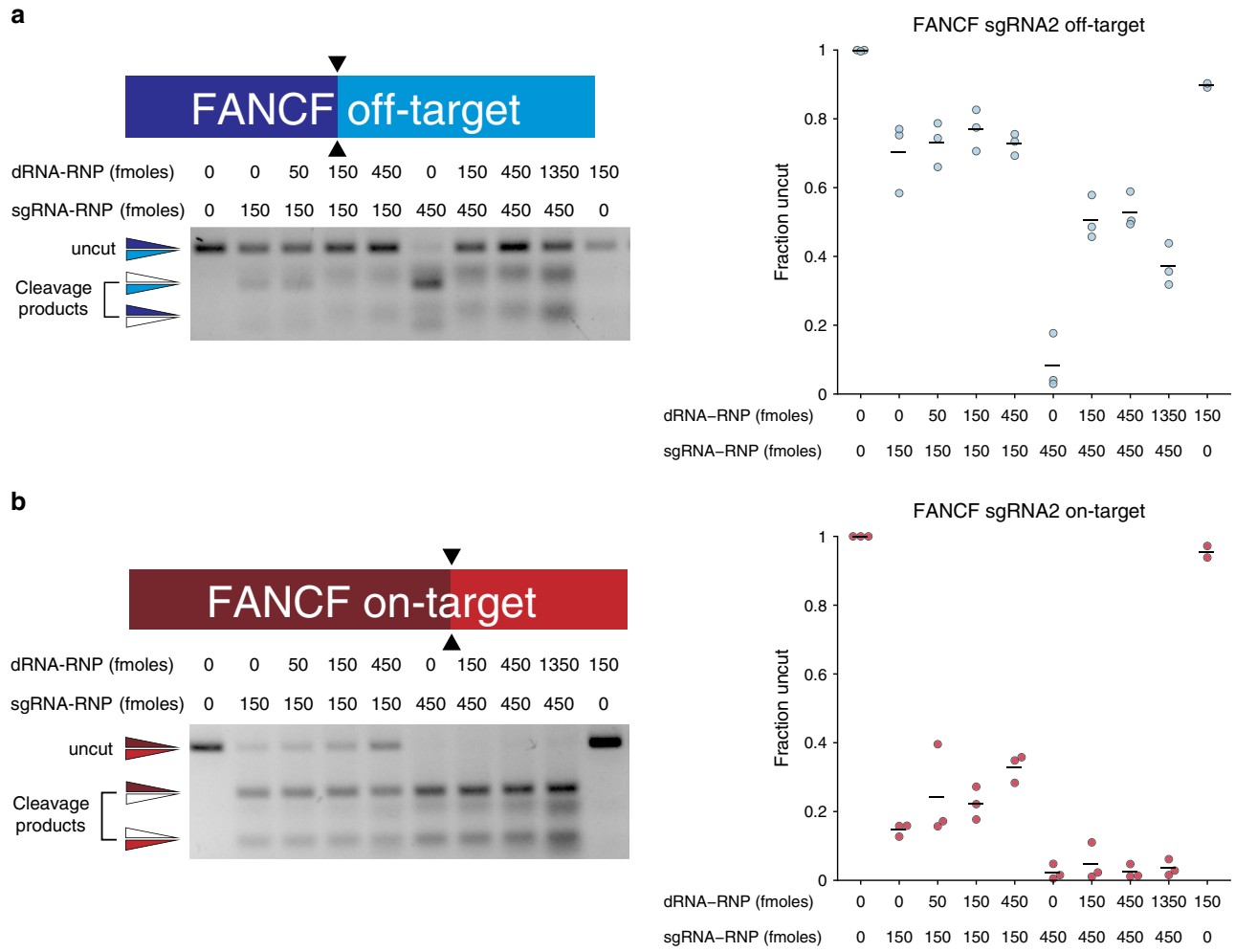

**Fig. 2 dRNAs suppress off-target editing by competing with sgRNAs for off-target sites.** Representative gels of in vitro cleavage of PCR products containing either **a** *FANCF*-sgRNA2 OT1 or **b** the *FANCF*-sgRNA2 on-target site with either 150 or 450 fmoles of Cas9 *FANCF*-sgRNA2 RNP in the presence of variable amounts of the Cas9 *FANCF*-sgRNA2 OT1 dRNA1 complex. Fraction of uncut DNA determined by gel densitometry. Means of $n = 3$ replicates depicted by solid lines. For uncropped gels, see Supplementary Fig. 14. Source data are available in the Source Data file.

Thus, we wondered whether they could be enhanced with dOTS. One approach uses truncated sgRNAs (tru-sgRNAs) with 17–19 base target sequences to increase on-target specificity at some loci. For example, truncation of the *VEGFA* sgRNA3 target sequence (*VEGFA* tru-sgRNA3) decreases editing at some off-target sites, but editing at OT2 remains[11]. dOTS suppressed editing at this refractory off-target site without affecting on-target editing (Fig. 4a), demonstrating that it is compatible with tru-sgRNAs.

More recently, rational engineering of SpCas9 has produced high-specificity variants like eSpCas9(1.1), SpCas9-HF1, and HypaCas9[18–20]. While these variants generally improve on-target specificity, they do not suppress unwanted editing at all off-target sites for all sgRNAs. For example, a recent evaluation of these three high-specificity variants revealed off-target editing by all three variants for four of the six sgRNAs tested[20]. In another example, *FANCF*-sgRNA2 OT1 is still edited at high frequencies by all three high-specificity variants (Fig. 4b)[18,20]. Co-transfection of *FANCF*-sgRNA2 with an effective dRNA reduced off-target editing to levels indistinguishable from non-transfected controls for all high-specificity Cas9 variants ($P > 0.05$, one-sided *t*-test, $n = 3$), dramatically increasing specificity ratios (Fig. 4b). dRNAs also effectively suppressed off-target editing by eSpCas9(1.1) and SpCas9-HF1 at a refractory *VEGFA* sgRNA3 off-target

(Supplementary Fig. 10). High-specificity Cas9 variants are known to exhibit decreased on-target activity, which is sensitive to delivery method and other factors[21,31,37]. Indeed, in some cases, we observe a decrease in on-target editing when high-specificity Cas9 variants and dOTS are combined. However, this reduction in on-target editing is generally less pronounced than the efficiency loss observed comparing HypCas9 or SpCas9-HF1 to wild-type in the absence of dOTS. The reduction in on-target editing is also markedly less than the degree of suppression achieved by dOTS at the off-target site. Thus, dOTS can be combined with many other methods for improving Cas9 specificity.

**dOTS can be multiplexed to suppress multiple off-targets.** Since many sgRNAs induce off-target editing at numerous sites[4,5,38], we examined whether dOTS could be multiplexed. We selected three off-target sites for *VEGFA* sgRNA2 with individually effective dRNAs (Fig. 1c, Supplementary Fig. 3). HEK-293T cells were transfected with *VEGFA* sgRNA2 and the dRNAs individually, in duplex, or in triplex. Even when all three dRNAs were combined, editing at each off-target site was suppressed by its cognate dRNA with only small losses in on-target editing (Fig. 5a; Supplementary Fig. 11a). Multiplexed dOTS was also effective for two other sgRNAs (Supplementary Fig. 11b, c), and

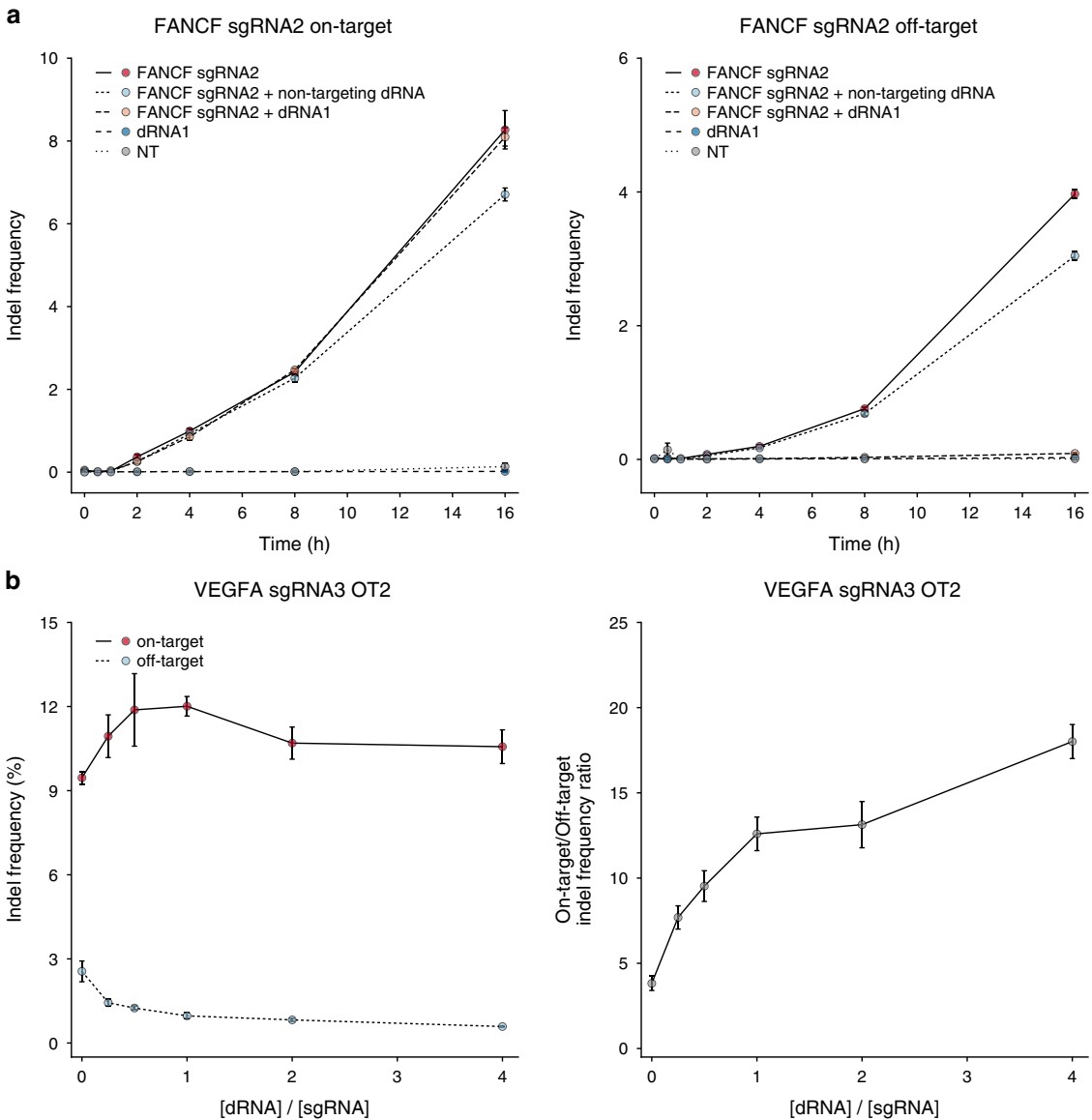

**Fig. 3 dRNAs affect off-target, but not on-target, editing kinetics and can be titrated to improve specificity. a** Editing of *FANCF*-sgRNA2 on-target and OT1 sites using chemically inducible Cas9 (ciCas9) from 0 to 16 h after activation with A115. Nontargeting dRNA is a 14-base control dRNA targeting a non-endogenous site. NT = non-transfected control. Points depict the mean of $n = 3$ biological replicates. Error bars show the standard error of the mean. **b** Indel frequencies and specificity ratios at *VEGFA* sgRNA3 on-target and OT2 sites in cells transfected with plasmids encoding Cas9 and varying ratios of *VEGFA* sgRNA3 and dRNA2. dRNA untreated cells were transfected with Cas9 and a 1:1 *VEGFA* sgRNA3:GFP plasmid ratio. Error bars depict s.e.m. ($n = 3$ cell culture replicates). OT off-target. Source data are available in the Source Data file.

could even suppress the off-targets of two distinct sgRNAs simultaneously (Supplementary Fig. 11d). Notably, each dRNA only impacted editing at its cognate off-target site, without increasing or decreasing the editing at the other off-target sites of the sgRNA.

Like wild-type Cas9, high-specificity Cas9 variants can cause editing at multiple off-target sites. For example, eSpCas9 reportedly drives appreciable editing with *VEGFA* sgRNA2 at three different off-target sites[20]. We observed off-target editing at two of these sites, and found that dRNAs could simultaneously decrease off-target editing at both sites without perturbing on-target editing (Fig. 5b). Furthermore, multiplexed dOTS suppressed editing driven by SpCas9-HF1 and HypaCas9 at these off-target sites (Supplementary Fig. 12). Thus, in the context of both wild-type and variant Cas9, dRNAs can be combined to suppress multiple off-targets simultaneously.

**dRNAs enable scarless HDR-mediated genome editing**. When mutations introduced by HDR do not substantially disrupt the target sequence or PAM, as is generally the case for single nucleotide variants, Cas9 can continue to cleave the target site after repair. Continued cleavage introduces indels, substantially decreasing the frequency of loci containing the desired sequence. For example, quantification of editing outcomes at *PSEN1* revealed that up to 95% of HDR-corrected templates showed secondary indels due to recutting[39]. If a protein-coding region is being edited, synonymous blocking mutations that disrupt the sgRNA target sequence, PAM, or both are generally included in the repair template. Unfortunately, synonymous blocking mutations may alter protein expression or interfere with mRNA splicing. Furthermore, predicting functionally neutral blocking mutations in non-coding regions is extremely challenging. Base editing can in some cases make single base changes, yet its use is

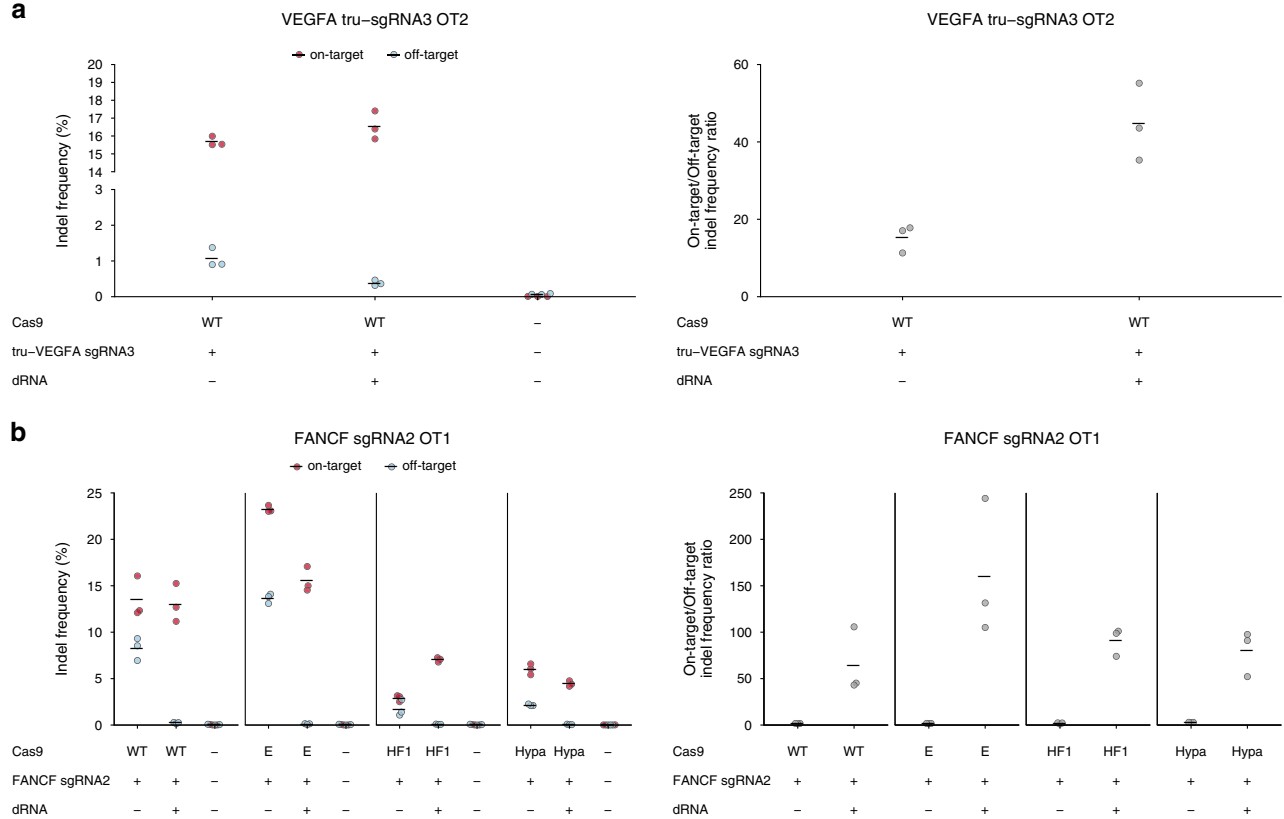

**Fig. 4 dRNAs can be combined with other approaches for improving Cas9 specificity.** Indel frequencies and specificity ratios 24 h after transfection with **a** plasmids encoding WT Cas9, a dRNA targeting *VEGFA* sgRNA3 OT2 (dRNA2) and a truncated guide *VEGFA* tru-sgRNA3, or **b** High-specificity variants of Cas9 and a dRNA targeting *FANCF*-sgRNA2 OT1 (dRNA1). Wild-type Cas9 (WT), eSpCas9 (E), SpCas9-HF1 (HF1), HypaCas9 (Hypa). Means of $n = 3$ cell culture replicates depicted by solid lines. OT off-target. Source data are available in the Source Data file.

hindered by unwanted bystander editing within the editing window, off-target editing of RNA, and an inability to install transversion mutations or targeted insertions and deletions[40–42]. Thus, scarless editing, the ability to efficiently introduce single nucleotide variants and other small changes into the genome via HDR without blocking mutations or unwanted indels, would be of tremendous utility.

We predicted that dRNAs directed at a desired, HDR-corrected sequence could shield repaired sites from recutting, an approach we call dRNA ReCutting Suppression (dReCS) (Fig. 6a). We evaluated the ability of dRNAs to improve the HDR-mediated conversion of BFP to GFP through substitution of a single amino acid. Previously, several blocking mutations were used to prevent recutting, yet only a single nucleotide change is needed to alter the His in BFP (CAT) to the Tyr in GFP (TAT)[43]. We selected a previously used sgRNA in which the permissive site within the PAM (i.e., N in NGG) for the BFP sgRNA corresponds to the mutated nucleotide. Thus, this sgRNA possesses perfect complementarity to both the native and HDR-repaired locus, representing a worst-case scenario in which Cas9•sgRNA is expected to efficiently recut HDR-repaired sites. HEK-293T cells with stably integrated BFP were transfected with a single stranded oligodeoxynucleotide (ssODN) donor template containing the single nucleotide change, the sgRNA targeting BFP, and one of three dRNAs with perfect complementarity to the GFP but not BFP sequence. After 4 days, in the absence of dRNA, scarless HDR conversion to GFP was inefficient, with 1.94% of cells expressing GFP by flow cytometry. In the presence of the best dRNA, absolute HDR efficiency increased to 3.77% (Fig. 6b; Supplementary Fig. 13), corresponding to an increase in the

percentage of all edited sites exhibiting scarless HDR from 9.53% (s.e.m. = 0.40, $n = 3$) to 19.72% (s.e.m. = 0.52, $n = 3$; Fig. 6c). Thus, dReCS can promote scarless HDR even when the sgRNA has perfect complementarity for the HDR-corrected sequence.

## Discussion

Here, we describe a general approach for the targeted suppression of unwanted Cas9-mediated editing that relies on coadministration of dRNAs with complementarity to the suppressed site. Our approach exploits the previously unappreciated phenomenon we refer to as Cas9 self-competition: the ability of different Cas9•guide RNA complexes to compete for a limited number of genomic target sites. We show that catalytically inactive Cas9, in this case Cas9 bound to a dRNA, can protect sites from undesired cleavage by active Cas9•sgRNA complexes. One application of this approach, dOTS, reduced editing at 15 distinct off-target sites, in some cases below the limit of detection by high-throughput sequencing. Another application, dReCS, facilitated the scarless introduction of a single base change that did not impact the PAM or target sequence. dReCS circumvents the need for blocking mutations, making it particularly useful for single nucleotide variants and small indels in non-coding regions of the genome where synonymous blocking mutations are not an option. In both cases, effective dRNAs can generally be rapidly identified with minimal screening. Moreover, dRNAs are effective in a variety of different cell lines and they can be combined to protect multiple off-target sites simultaneously.

dOTS and dReCS offer many advantages, but they are not perfect. We could not find an effective dRNA for four of the 19

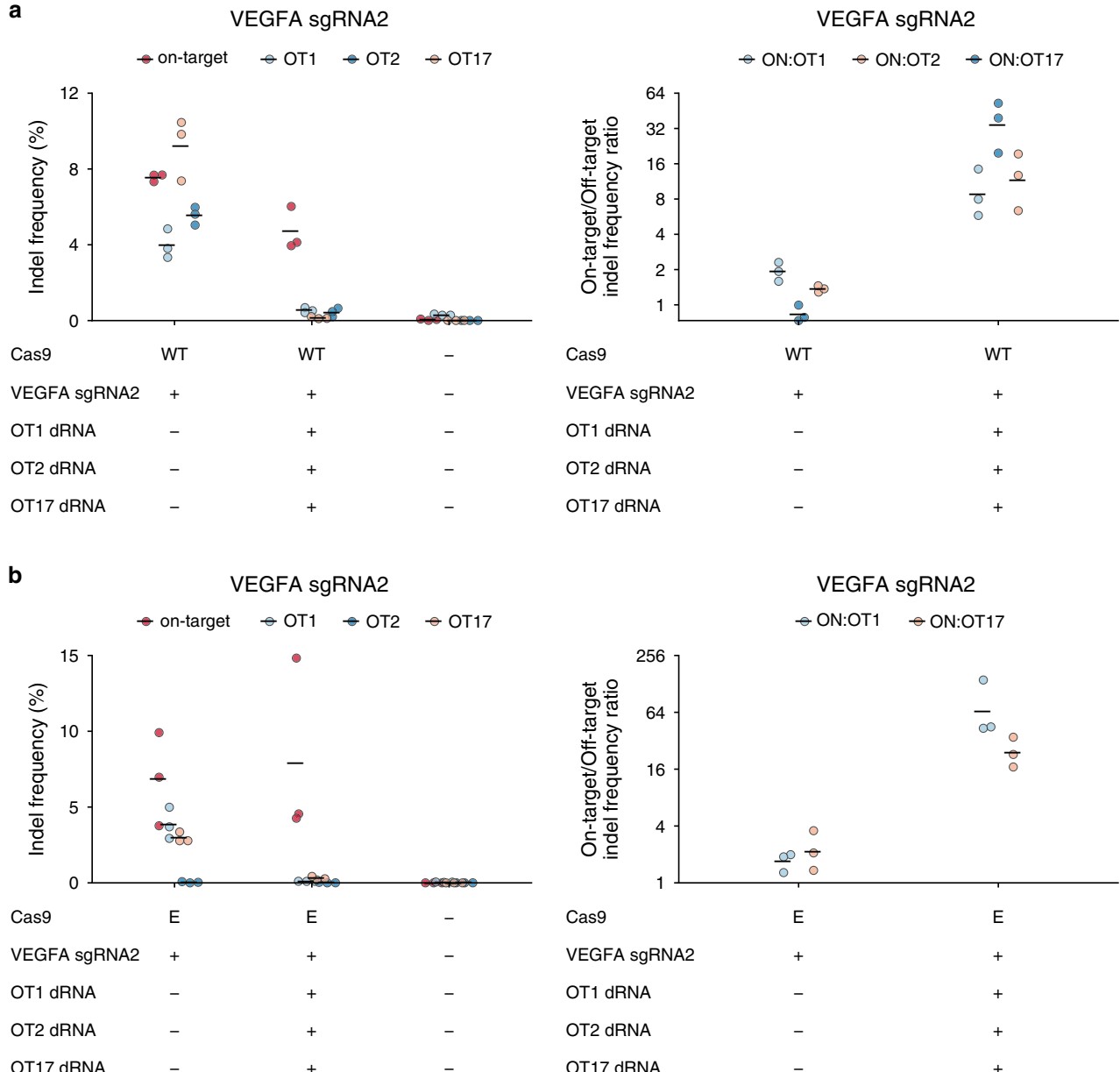

**Fig. 5 dRNAs can be multiplexed to suppress several off-targets simultaneously.** Indel frequencies and specificity ratios 24 h after transfection of plasmids encoding either **a** wild-type (WT) or **b** eSpCas9 (E), *VEGFA* sgRNA2, and dRNAs targeting one of three *VEGFA* sgRNA2 off-targets (OT1 dRNA1, OT2 dRNA8, OT17 dRNA8). Means of *n* = 3 cell culture replicates depicted by solid lines. OT off-target. Source data are available in the Source Data file.

target/off-target pairs we tested. In some cases, additional dRNAs could be screened, but the sequence restrictions imposed by the SpCas9 NGG PAM mean that effective dRNAs may not always exist. One alternative is to improve poorly performing dRNAs by manipulating dRNA/sgRNA ratios. Another is to combine dRNAs with the recently described xCas9 or SpCas9-NG variants, which have a more permissive PAM that increases the number of candidate dRNAs[44,45]. Another drawback is that some dRNAs decrease on-target editing, particularly when they are multiplexed to suppress several off-target sites simultaneously. We suspect that these losses in on-target editing likely arise due to dilution of the plasmids or competition between sgRNAs and dRNAs to complex with Cas9. The first issue could be addressed by using a multiplex guide expression scheme[46,47], and both could be addressed by delivering preformed ribonucleoprotein (RNP) mixtures[48]. Finally, dRNAs could yield unwanted transcriptional off-target effects. However, transcriptional repression by Cas9 in

the absence of a repressive domain is modest[49,50], and such effects would be transient unless both Cas9 and the dRNA were integrated into the genome.

Other approaches for minimizing off-target editing are also imperfect, as they reduce on-target efficiency[6–9,21,22], introduce new off-target sites[11,14,15], limit the number of potential target sites[11,14–17], or demand difficult Cas9 engineering[18–22,51,52]. Moreover, many of these approaches are laborious to implement in experimental models where Cas9 or a variant thereof has already been stably integrated into the genome[6–9,16–22,51,52]. Finally, these existing methods are generally incompatible with each other, meaning they cannot be used in concert to minimize limitations and improve performance. In contrast, dOTS and dReCS are comparatively easy to use, low-cost, and flexible. For example, dOTS could be used to address refractory off-targets of the popular engineered high-specificity Cas9 variants[18–22,51,52]. Here, we showed that dOTS could effectively suppress editing at

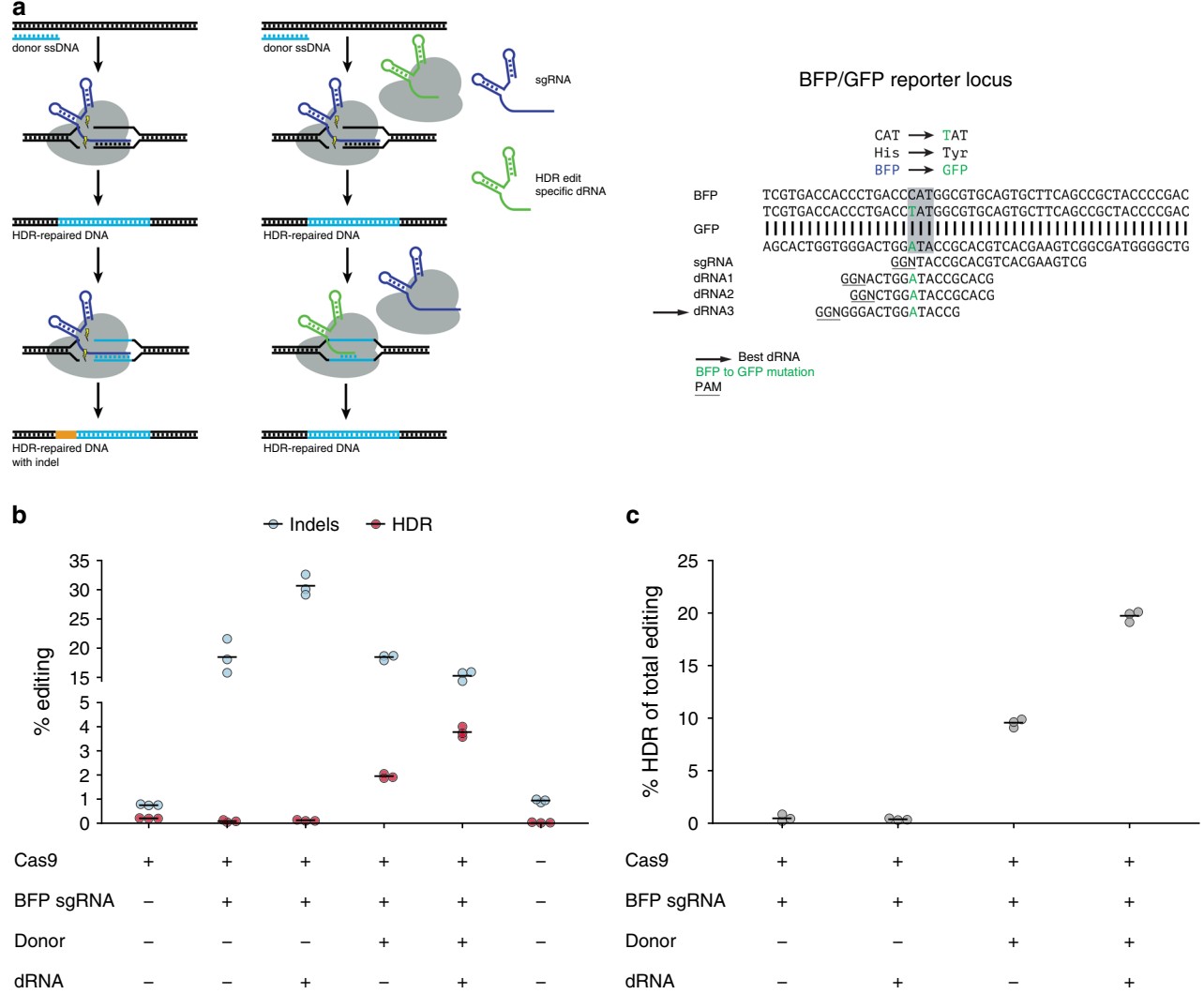

**Fig. 6 dRNA on-target recutting suppression (dReCS) facilitates scarless HDR. a** Schematic depicting dReCS and alignment of BFP, GFP, sgRNA, and dRNA sequences. dRNA (green) exhibiting perfect complementarity for the repaired site directs Cas9 binding but not cleavage, protecting the repaired site. Single base change to generate GFP from BFP is displayed in green with affected codon indicated by gray box. PAM sequences are underlined. Black arrow indicates best dRNA, as determined by maximal improvement in HDR yield. **b** Indels and homology-directed repair (HDR) as assessed by flow cytometry, where indels lead to a loss of BFP signal, and HDR leads to a loss of BFP and gain of GFP signal. **c** HDR as a percentage of total Cas9 edits observed. Means of $n = 3$ cell culture replicates depicted by solid lines. dRNA BFP sgRNA1 dRNA3 (see Supplementary Fig. 13, Supplementary Data Set 1). Source data are available in the Source Data file.

four refractory off-target sites with three high-specificity Cas9 variants. Using dOTS to address these refractory off-targets is also far less laborious and time-intensive than further Cas9 engineering, as has been done previously[18,51]. Additionally, dReCS is simpler and less time-consuming than CORRECT[39], a previous approach for scarless HDR editing that requires multiple rounds of HDR to introduce and subsequently remove blocking mutations. Because of their flexibility and technical simplicity, dOTS and dReCS could be readily integrated with existing protocols and experimental systems, enabling refinement of genome editing with minimal effort.

The flexibility of dOTS and dReCS means that they have applications beyond those we demonstrated. For instance, dOTS could facilitate allele-specific editing, even when the two alleles cannot be distinguished by a Cas9•sgRNA complex alone. Based on the principle of Cas9 self-competition, electroporation of Cas9•dRNA RNPs to quench editing by the active Cas9•sgRNA RNP should allow fine tuning of editing efficiencies. Similarly, dOTS could be employed to modulate the editing rates in CRISPR

lineage tracing[53]. Finally, dOTS and dReCS are likely to be effective with other CRISPR enzymes, such as SaCas9 or Cpf1. Thus, dOTS and dReCS are easy-to-implement, effective and complementary methods for refining genome editing in both research and clinical applications.

## Methods

**Expression plasmids**. All sgRNA and dRNA target sequences, except for *VEGFA* sgRNAs, were cloned into the gRNA_Cloning Vector according to the hCRISPR gRNA synthesis protocol (https://www.addgene.org/static/data/93/40/adf4a4fe-5e77-11e2-9c30-003048dd6500.pdf). gRNA_Cloning Vector was a gift from G. Church, Harvard (Addgene plasmid 41824). *VEGFA* site#1 ('*VEGFA* sgRNA1'), *VEGFA* site#2 ('*VEGFA* sgRNA2'), and *VEGFA* Site#3 ('*VEGFA* sgRNA3') were gifts from K. Joung, Massachusetts General Hospital (Addgene plasmids 47505, 47506, and 47507).

An N-terminal FLAG tag sequence was appended via Gibson Assembly Cloning (New England Biosciences) to a human codon optimized Cas9 (subcloned from hCas9, a gift from G. Church, Harvard; Addgene plasmid 41815) with a single C-terminal NLS expressed from a pcDNA3.3-TOPO vector. This was subsequently cloned into the pcDNA5/FRT/TO backbone (ThermoFisher). High-specificity variants of Cas9 — eSpCas9(1.1) (gift from F. Zheng, Broad Institute; Addgene

plasmid 71814) and VP12 ('SpCas9-HF1', gift from K. Joung, Massachusetts General Hospital; Addgene plasmid 72247) were subcloned into pcDNA5/FRT/TO backbone (ThermoFisher). HypaCas9 ('BPK4410') was a gift from J. Doudna and K. Joung, University of California, Berkeley and Massachusetts General Hospital (Addgene plasmid 101178).

The sequences of all plasmids, primers, and other DNA constructs used in this work can be found in Supplementary Data Set 1.

**dRNA design.** dRNA sequences were designed by identifying 14–16 nucleotide dRNA spacer sequences which met the following criteria: (1) the dRNA spacer sequence and/or its PAM overlaps with the off-target spacer sequence and/or its PAM, and (2) the dRNA spacer or PAM exhibits perfect complementarity to the off-target but not the on-target locus. Spacer sequences with a 5' G were preferentially selected, but spacers containing a mismatched 5' G were also used. Exhaustive screening of all candidate dRNAs, which met the criteria was not performed at all sites. Alignments of the on-target sites, off-target sites, and dRNAs used in this study are presented in Supplementary Fig. 4.

**Cell culture.** HEK-293T cells (293T/17, ATCC) were maintained in high-glucose DMEM supplemented with 10% fetal bovine serum (FBS, Life Technologies). U2OS cells (ATCC) were maintained in McCoy's 5A (modified) medium supplemented with 10% FBS (Life Technologies). hESC Elf1 iCas9[27] were plated into matrigel-coated 24-well plates and cultured in MEF-conditioned media supplemented with 2iL-I-F (GSK3i, MEKi, LIF, IGF, bFGF). All cell lines were regularly tested and confirmed free from mycoplasma contamination.

**Genome editing by Cas9.** Unless otherwise specified, HEK-293T cells were plated in 24-well plates at $1.5 \times 10^5$ cells/well. The day after plating, cells were transfected with Turbofectin 8.0 (Origene). For all dOTS experiments, 1.5 μL of Turbofectin 8.0 and 500 ng of plasmid DNA were transfected. For dRNA screening experiments, the plasmid DNA mixture contained 250 ng Cas9 (eSpCas9, Cas9-HF1, or HypaCas9), 125 ng sgRNA, and 125 ng dRNA. For wells without dRNA, the 125 ng of pMAX-GFP was substituted for the dRNA plasmid as a transfection control. For multiplex dOTS experiments, the plasmid DNA mixture contained 250 ng Cas9, 125 ng sgRNA, and 125 ng each of 1–3 dRNAs. A pMAX-GFP plasmid was used to increase total DNA transfected per well to 750 ng. U2OS cells were plated in 12-well plates at $7.5 \times 10^4$ cells/well. The next day they were transfected with 3 μL of Turbofectin 8.0 and a total of 1 μg plasmid DNA (500 ng Cas9, 250 ng sgRNA, and 250 ng dRNA or pMAX-GFP plasmid). For titration experiments with all sgRNAs except *VEGFA* sgRNA3, HEK-293T cells were transfected with 1.5 μL of Turbofectin 8.0 and 500 ng of plasmid DNA. This DNA mixture contained 250 ng Cas9. The remaining 250 ng of DNA was divided between sgRNA and dRNA at varying ratios such that the total DNA was kept constant across experiments (1:1, 125 ng each sgRNA and dRNA; 1:2, 83.3 ng sgRNA and 166.7 ng dRNA; 1:4, 50 ng sgRNA and 200 ng dRNA; 2:1, 166.7 ng sgRNA and 83.3 ng dRNA; and 4:1, 200 ng sgRNA and 50 ng dRNA). For wells without dRNA, 125 ng of pMAX-GFP plasmid was substituted for the dRNA plasmid as a transfection control. For titration experiments with *VEGFA* sgRNA3, HEK-293T cells were transfected as above, but the DNA mixture contained 166.5 ng Cas9, and the various sgRNA:dRNA ratios were as follows (1:1, 166.5 ng each sgRNA and dRNA; 1:2, 111 ng sgRNA and 222 dRNA; 1:4, 66.6 ng sgRNA and 266.4 ng dRNA; 2:1, 222 ng sgRNA and 111 ng dRNA; 4:1, 266.4 ng sgRNA and 66.4 ng dRNA). For wells without dRNA, 166.5 ng of pMAX-GFP plasmid was substituted for the dRNA plasmid as a transfection control.

To harvest HEK-293T and U2OS cells for dOTS experiments, 24 h after transfection each well of a 24-well plate was resuspended by thorough pipetting with 400 μL ice-cold DPBS. Resuspended cells were then spun at $1500 \times g$ for 10 min at 4 °C. DPBS was then aspirated and cell pellets were stored at −80 °C until genomic DNA isolation. For extended timepoint experiments, the same protocol was followed, except cells were passaged into a new 24-well plate after 24 h after transfection and then subsequently harvested 48 h after passaging.

Two days prior to plating, hESC Elf1 iCas9 cells were treated with 2 μg/ml doxycycline to induce Cas9 expression. At day 0, $2.5 \times 10^4$ cells were plated into each well of a 24-well plate with addition of fresh doxycycline (2 μg/ml) and 10 μM Rock inhibitor to promote cell survival. After 24 h, cells were transfected with 3 μL of Genejuice (EMD Millipore) and 1 μg plasmid DNA. This plasmid DNA mixture contained 500 ng sgRNA and 500 ng dRNA. For wells without dRNA, 500 ng of pMAX-GFP was substituted as a transfection control.

For Elf1 cells, 48 h after transfection, each well of a 24-well plate was rinsed once with 0.5 mL DPBS and incubated for 5 min with trypsin to detach cells. 5 mL hESC media was added and the cells were spun down at $290 \times g$ for 3 min. The pellet was then washed with 1 mL DPBS, spun again at $290 \times g$ for 3 min then flash frozen in liquid nitrogen and stored at −80 °C until genomic DNA isolation.

For GUIDE-seq experiments, U2OS cells were electroporated following previously established protocols[5,20]. Briefly, $2 \times 10^5$ cells per condition were transfected with 500 ng Cas9 plasmid, 250 ng sgRNA plasmid, 250 ng dRNA plasmid, and 100 pmol of an end-protected double-stranded oligonucleotide (dsODN) GUIDE-seq tag. For wells without dRNA or sgRNA, pMAX-GFP plasmid was substituted as a transfection control. Twenty microliter transfections

were performed using a Lonza 4D nucleofector X unit and SE kit using the DN-100 program. Cells were replated in 96-well plates after transfection and harvested for genomic DNA 96 h later.

**dReCS.** For dReCS experiments, a HEK-293T cell line with a genomically encoded BFP/GFP reporter was used[43]. The BFP/GFP reporter HEK-293T cell line contains a BFP that is converted to GFP via HDR-mediated substitution of a single amino acid (His in BFP (C**A**T) to Tyr in GFP (**T**AT)). BFP/GFP reporter cells were plated at $3.0 \times 10^5$ cells/well in 12-well plates. 18 h after plating, cells were transfected with 3 μL of Turbofectin 8.0 (Origene) and 1000 ng of total DNA. The total DNA mixture contained 272.7 ng of plasmid encoding Cas9, 54.5 ng sgRNA plasmid, 218 ng dRNA plasmid, and 454.5 ng symmetric or asymmetric single stranded donor DNA (Supplementary Data Set 1)[43]. For controls missing one or more of these DNA elements, the appropriate amount of DNA was replaced with a pKan-mCherry plasmid. Cells were maintained with standard passaging procedures for 4 days post-transfection until analysis by flow cytometry.

After 4 days, cells were washed with 2 mL DPBS, trypsinized with 0.5 mL 0.25% trypsin-EDTA (Life Technologies) for 2–4 min, and quenched with DMEM supplemented with 10% FBS. Cells were then spun down at $290 \times g$ for 4 min, aspirated, and resuspended in DPBS supplemented with 1% FBS. Cells were run through a 35 μm filter and analyzed by flow cytometry on an LSR-II flow cytometer. After gating for live cells (FSC-A vs SSC-A) and single cells (FSC-A × SSC-W), cells were analyzed for their BFP and GFP fluorescence. Gates for BFP and GFP positivity were determined by comparison to an untransfected BFP cell line. BFP+ GFP− cells were considered wild-type (WT). BFP− GFP− cells were considered to have undergone NHEJ but not HDR, as indels in this region of BFP lead to loss of fluorescence. Any cell that was GFP+ (regardless of residual BFP fluorescence) was considered to have undergone successful HDR. Percentages for each result (WT, HDR, and NHEJ) were calculated as a fraction of the total cells that passed singlet gating. Percent HDR of total editing was determined as the fraction of cells with successful HDR divided by the total number of cells that underwent either HDR or NHEJ.

**In vitro Cas9-RNP nuclease assays.** Cas9-2NLS in a pMJ915 vector (Addgene plasmid 69090) was expressed in *E. coli* and purified by a combination of affinity, ion exchange, and size exclusion chromatography as previously described[54], except the final purified protein was eluted into a buffer containing 20 mM HEPES KOH pH 7.5, 5% glycerol, 150 mM KCl, 1 mM DTT at a final concentration of 40 μM of Cas9-2NLS. *FANCF*-sgRNA2 and *FANCF* dRNA1 were generated by HiScribe (NEB E2050S) T7 in vitro transcription using PCR-generated DNA as a template[54], (https://doi.org/10.17504/protocols.io.dm749m). Complete sequences for all sgRNA templates can be found in Supplementary Data Set 1.

A 463 basepair fragment containing the on-target cut site of *FANCF*-sgRNA2 (*FANCF* target site) was PCR amplified from a custom *FANCF*-sgRNA2 target site substrate gBlock (IDT) using primers oCR1711 and oCR1712. A 329 basepair fragment containing the cut site for off-target 1 of *FANCF*-sgRNA2 (*FANCF* off-target) was PCR amplified from a custom *FANCF*-sgRNA2 off-target substrate gBlock (IDT) using oCR1713 and oCR1714 (Supplementary Data Set 1). Prior to nuclease experiments, sgRNA and dRNA-RNP complexes were generated by incubating purified Cas9-2NLS and *FANCF*-sgRNA2 or dRNA1 in equimolar amounts for 10 min. For dRNA-RNP titration experiments, 150 or 450 fmoles of *FANCF*-sgRNA2-RNP complex and 0, 50, 150, or 450 fmoles of dRNA-RNP Cas9-sgRNA complex were coadded to 150 fmoles of *FANCF* target site or *FANCF* off-target substrate DNA. Reaction mixtures were incubated at 37 °C for 20 min in 20 mM Tris, 100 mM KCl, 5 mM MgCl₂, 1 mM DTT, 0.01% Tween, 50 μg/mL Heparin. Reactions were stopped by the addition of 1:4 volume of STOP solution (8 mM Tris, 0.025% BPB, 0.025% XC, 50% Glycerol, 110 mM EDTA, 1% SDS, 3 mg/mL Proteinase K), followed by incubation at 55 °C for 5 min to liberate cut DNA fragments. Each digestion reaction was run on a 2% TAE agarose gel, post-stained with Ethidium Bromide, and resolved on a Gel-Doc (BioRad).

For preincubation experiments, *FANCF*-sgRNA2 or dRNA1 RNP complexes were generated as described above. 450 fmoles of a single RNP complex was added to 150 fmoles of *FANCF* target site or *FANCF* off-target substrate DNA and incubated at 37 °C for 10 min. After 10 min, 450 fmoles of the other Cas9-RNP complex was added and allowed to incubate at 37 °C for an additional 10 min. Reactions were quenched, incubated, and run on a gel in an identical manner to the above experiments.

Gel densitometry analysis was performed in ImageJ. For each lane, background density was subtracted from the quantification of each band. The density of the uncut band was then divided by the total intensity of all bands in the lane to determine the uncut DNA fraction.

**Genomic editing by ciCas9.** HEK-293T cells were treated according to previous methods[6]. Briefly, HEK-293T cells were plated in 12-well plates at $3.0 \times 10^5$ cells/well. The day after plating, cells were transfected with 1.5 μL Turbofectin 8.0 and 500 ng of plasmid DNA. The plasmid DNA mixture contained 250 ng Cas9, 125 ng *FANCF*-sgRNA2 sgRNA, and 125 ng dRNA. For wells without dRNA, the 125 ng of dRNA plasmid were replaced by pMAX-GFP as a transfection control.

Twenty-four hours after transfection, cells were treated with with 10 μM A115 dissolved in DMSO to induce ciCas9 activity. 24 h after treatment with A115, cells were harvested after washing with 600 μL DPBS to remove excess A115 and then resuspending cells in 600 μL ice-cold DPBS. Resuspended cells were then spun at 1500 × g for 10 min at 4 °C. DPBS was aspirated and the cell pellets were stored at −80 °C until genomic DNA isolation.

**Indel detection by high-throughput sequencing.** Genomic DNA isolation, sequencing, and analysis were performed as previously described[6]. Briefly, genomic DNA was isolated using the DNEasy Blood and Tissue Kit (Qiagen) according to the manufacturer's instructions except that the proteinase K digestion was conducted for 1 h at 56 °C. Fifteen cycles of primary PCR to amplify the region of interest was performed using 2 μL of DNeasy eluate (~100–300 ng template) in a 5 μL Kapa HiFi HotStart polymerase reaction (Kapa Biosystems; for primers see Supplementary Data Set 1). The PCR reaction was diluted with 35 μL DNAse-free water (Ambion). Illumina adapters and indexing sequences were added via 20 cycles of secondary PCR with 3 μL of diluted primary PCR product in a 10 μL Kapa Robust HotStart polymerase reaction (New England Biosciences; for primers see Supplementary Data Set 1). The final amplicons were run on a TBE-agarose gel (1.5%); and the product band was excised and extracted using the Freeze and Squeeze Kit according to the manufacturer's instructions (Bio-Rad). Gel-purified amplicons were quantified using Qbit dsDNA HS Assay kit (Invitrogen). Then, up to 1200 indexed amplicons were pooled, quantified by Kapa Library Quantification (Kapa Biosystems) and sequenced on a NextSeq (NextSeq 150/300 Mid V2 kit, Illumina, for primers see Supplementary Data Set 1).

Indels were quantified as previously described[6]. Briefly, after demultiplexing of reads (bcl2fastq/2.18, Illumina), indels were quantified with a custom Python script that is freely available upon request. 8-mer sequences were identified in the reference sequence located 20 bp upstream and downstream of the target sequence. Sequence distal to these 8-mers was trimmed. Reads lacking these 8-mers were discarded. For the VEGFA sgRNA3 OT2 locus, the process was the same, except 20-mer sequences located 10 bp upstream and downstream of the target sequence were used. For the VEGFA sgRNA3 OT4 locus, 8-mer sequences located 10 bp upstream and downstream of the target sequence were used. The trimmed reads were then evaluated for indels using the Python difflib package. Indels were defined as trimmed reads, which differed in length from the trimmed reference and for which an insertion or deletion operation spanning or within 1 bp of the predicted Cas9 cleavage site was present. For dRNA only experiments, indels were quantified using both the sgRNA and dRNA predicted cut sites. Specificity ratios were calculated by dividing the indel percentage at the on-target locus by the indel percentage at the off-target locus for each sgRNA. For quantification of off-target editing for one of the VEGFA tru-sgRNA3 plus dRNA replicates (Fig. 4a), reads were acquired from multiple sequencing runs.

**GUIDE-seq.** Calculation of indels was performed at the FANCF-sgRNA2 ON and OT1 loci as described above. To determine the percentage of reads containing a dsODN tag, the same Python script as above was used and modified to count integration of the full length dsODN within 1 bp of the predicted Cas9 cleavage site. A ratio of dsODN-containing reads to indel-containing reads was calculated. To perform GUIDE-seq analysis, samples were prepared according to established protocols[5]. Briefly, genomic DNA was isolated using the DNEasy Blood and Tissue Kit (Qiagen) according to manufacturer's instructions except that the proteinase K digestion was conducted for 1 h at 56 °C. DNA was sheared using a Covaris LE220 to an average size of 500 bp and cleaned using Ampure XPRI beads according to the manufacturer's protocol. DNA was then end-repaired, A-tailed, and ligated to adapters containing an 8 nt unique molecular identifier. Samples were then amplified with two rounds of nested PCR with primers that complement the oligo tag. Sample libraries were prepared as described above and sequenced on an Illumina MiSeq. Data were analyzed with the GUIDE-seq software[55] allowing for up to eight mismatches with a modification of a 35 bp window for detected off-target alignments to the reference sequence. Frequency of dsODN-containing reads genome-wide were calculated per sample.

**Statistical analysis.** Statistical analysis of indel frequency and specificity ratios were performed using a one-sided two sample Student's t-test.

**Reporting summary.** Further information on research design is available in the Nature Research Reporting Summary linked to this article.

## Data availability
Raw sequencing data have been uploaded to the SRA with BioProject accession number PRJNA629634. The source data for Figs. 1–6 and the Supplementary Figures are available in the Source Data file. All other data are available from the authors upon reasonable request.

## Code availability
Custom python scripts for indel quantitation and R scripts for figure generation are available on github.

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

## Acknowledgements

This work was supported by the NIH (R01GM086858 (D.J.M.), R01GM109110 (D.M.F.), F30CA189793 (J.C.R.)), and NSF (0954242 (D.J.M.)). D.M.F. is a CIFAR Azrieli Global Scholar. We would like to thank D. Prunkard for assistance with flow cytometry experiments and analysis, as well as C. Murray for advice and assistance in the conception of this project.

## Author contributions

J.C.R. conceived the study. J.C.R., N.A.P., J.E.C., D.M.F., and D.J.M., designed the experiments. J.C.R., N.A.P., C.D.R., J.M., C.T.W., and J.J.S. performed the experiments. J.J.S. prepared samples for high-throughput sequencing. J.C.R., N.A.P., D.M.F., and D.J.M. wrote and edited the manuscript. All authors approved the final manuscript.

## Competing interests

The authors declare no competing interests.
