## [Peer Review File · Nature Communications]

Reviewers' Comments:

Reviewer #1:

Remarks to the Author:

The authors have addressed my concerns.

Reviewer #2:

Remarks to the Author:

Rose et al. describe how the co-administration of truncated guide-RNA (dRNA) corresponding to the known off-target sites can reduce the off-target activity of SpCas9:sgRNA. The dOTS method is based on a simple hypothesis, and it is working well for suppressing off-target editing at multiple genomic loci. In particular, multiplexing is possible to block multiple off-target sites simultaneously. Moreover, it is easy to use in many laboratories because the only thing required is to express extra dRNAs. Finally, dReCS method would be a useful tool for scarless precision genome editing.

This paper is well-written with sufficient explanations for their hypothesis and experimental results. I think the dOTS method is worth to be reported in the Nature Communications. However, there are several issues that should be addressed, mostly at the discussion part.

1. One of the major concerns that I have, the authors missed the big picture of the hypothesis. The work focused on some biased kinds of off-target events without any explanation. Authors should have discussed all kinds of off-target events that arise from partial spacer complementarity at other non-target loci, for example, number of mismatches, the position of the mismatch and the combination of the two. It's quite apparent that the present system will work only for the off-targets due to sequence mismatch in the PAM or seeding region of the spacer. The authors should discuss and investigate the off-target activity due to mismatch at the distal end of the spacer.

2. Another major concern is the generalizability of the approach. The method will work only when we have information on the genome-wide off-target activity. That too, not all kinds of off-targets can be suppressed. Moreover, multiple off-targets might be difficult as it will require co-administer of multiple dRNAs, which will compete with the sgRNA. Titration of different guides or preparing a cocktail of RNPs seems too laborious.

3. Is there any rule to choose the candidate dRNAs? Did the authors test all potential dRNAs as long as PAM sequences are available? How does the distance between the off-target cleavage site and dRNA binding site affect the performance of dRNAs? How did the authors pick the length of dRNA spacers among 14 to 16 nt? It would be good if the authors give some guidelines for designing the best dRNAs.

4. Fig 2. Why dRNA RNP 1350 fmoles shows higher OT cleavage than lower concentrations? The unit is missing on the right plots for both a and b panels.

5. Fig 4. Y-axes need to be properly labeled. The reduction in the on-target activity in the case of eSpCas9 needs to be explained by considering the fact that lower DNA binding affinity of eSpCas9 can alter competition between sgRNA and dRNA RNPs.

6. Fig 5. Error bar in the panel b is beyond acceptable value, needs to be replaced.

7. SI, Fig 2, panel d: results indicate that the introduction of dRNA1 results in a complex situation. Apart from reducing OT1 activity and enhancing on-target activity it also increased off-target activity at other sites. Authors need to discuss the observation and their implication on the dRNA approach in a broader perspective.

8. Blocking the recutting of the HDR-repaired gene by dRNA is a good strategy, however, its application is limited by the narrow scope of the approach itself. The authors need to present a diagram explaining the gene target, HDR-repaired (BFP to GFP) gene sequence and dRNA alignment with the repaired product for a better understanding. While the author reports mutation from CAT to TAT codon for BFP to GFP editing, however, the mentioned reference (Nature Biotechnology, 2016, 34, 339–344) suggests changing to TAC codon instead. In addition, reference work also suggests mutation in the PAM sequence to impair recutting. The authors need to present the design clearly.

9. Supplementary figures 6 and 7 didn't appear on the manuscript.

10. It seems like that fine-tuning the expression level of gRNA and dRNA is important for maximizing the on-target editing and minimizing off-target indels. This would be easier when Cas9 is delivered in the form of a ribonucleoprotein (RNP) complex. Moreover, RNP delivery is a better option for reducing off-target effects. In fact, the authors already described that RNP delivery could be an alternative for keeping the on-target activity in multiplexing and for the allele-specific editing. So, it would be great if the authors can present data from RNP delivery-based dOTS method, but it's not a must.

Response to Reviewer Comments

In this document, we provide a point-by-point response to each reviewer concern. Referee comments are shown in italics. Where applicable, we cite changes made by page, paragraph and figure number.

REVIEWERS' COMMENTS:

Reviewer #1 (Remarks to the Author):

The authors have addressed my concerns.

Thank you, again, for taking the time to review our work.

Reviewer #2 (Remarks to the Author):

Rose et al. describe how the co-administration of truncated guide-RNA (dRNA) corresponding to the known off-target sites can reduce the off-target activity of SpCas9:sgRNA. The dOTS method is based on a simple hypothesis, and it is working well for suppressing off-target editing at multiple genomic loci. In particular, multiplexing is possible to block multiple off-target sites simultaneously. Moreover, it is easy to use in many laboratories because the only thing required is to express extra dRNAs. Finally, dReCS method would be a useful tool for scarless precision genome editing.

This paper is well-written with sufficient explanations for their hypothesis and experimental results. I think the dOTS method is worth to be reported in the Nature Communications. However, there are several issues that should be addressed, mostly at the discussion part.

1. One of the major concerns that I have, the authors missed the big picture of the hypothesis. The work focused on some biased kinds of off-target events without any explanation. Authors should have discussed all kinds of off-target events that arise from partial spacer complementarity at other non-target loci, for example, number of mismatches, the position of the mismatch and the combination of the two. It's quite apparent that the present system will work only for the off-targets due to sequence mismatch in the PAM or seeding region of the spacer. The authors should discuss and investigate the off-target activity due to mismatch at the distal end of the spacer.

- We appreciate the reviewer's concern, but it is factually incorrect. dOTS is not limited to off-targets where mismatches to the sgRNA lie in the seed sequence. In this study, we demonstrated suppression of numerous off-targets with no mismatches in the PAM proximal seed sequence (VEGFA sgRNA1 OT1, VEGFA sgRNA1 OT4, VEGFA sgRNA2 OT2 & VEGFA sgRNA2 OT17, and VEGFA sgRNA3 OT18; **Fig. 1c & S3**). The 19 off-targets investigated in this study represent diverse kinds of off-targets, including off-targets with single or multiple

mismatches, mismatches in the seed region or PAM distal segment of the spacer. We have included a new supplemental figure (**Supplementary Figure 4**) with alignments of on-and off-targets as well as dRNAs to better illustrate the types of off-targets included in this study, as well as the relative positioning and sequences of the dRNAs used.

We did not observe any patterns in terms of mismatch number or position that were predictive of dOTS efficacy. Our ability to detect such patterns was limited in large part by the small number of sites for which we were not able to identify an effective dRNA. Prior to our initial submission to *Nature Communications*, we became aware of another manuscript describing an approach similar to dOTS. This manuscript was submitted along with ours. That manuscript examines additional on/off-target pairs in order to discern design rules for suppressing off-targets with dRNAs.

We are unclear as to what the reviewer is referring to when they discuss a mismatch in the PAM. There are some off-targets in which the permissive position (i.e. the “N” in the NGG PAM) differs between the on- and off-target sites. However, it is not appropriate to describe this as a mismatch. Variation at this position is essentially invisible to the Cas•sgRNA complex. To our knowledge, our approach is the only that can decrease unwanted editing at a site in which the only differential position relative to the on-target corresponds to the permissive PAM position. As evidence of this, dReCS was able to protect an HDR-edited sequence that only differed from the original target sequence at the N position of the NGG PAM (**Fig. 6**).

2. Another major concern is the generalizability of the approach. The method will work only when we have information on the genome-wide off-target activity. That too, not all kinds of off-targets can be suppressed. Moreover, multiple off-targets might be difficult as it will require co-administer of multiple dRNAs, which will compete with the sgRNA. Titration of different guides or preparing a cocktail of RNPs seems too laborious.

- As discussed in the manuscript, dOTS is not intended to replace approaches such as the use of engineered high-specificity Cas9 variants. Rather, it is meant to selectively suppress known off-targets, such as those sites that are not suppressed by the use of high-specificity variants.
 - “All high-specificity Cas9 variants appear to balance on- vs off-target activity via the same mechanism^{20,23} and, as a consequence, often fail to suppress editing at the same obstinate off-target sites^{20,22}. Thus, new methods for off-target suppression are needed, particularly ones that preserve on-target editing, can be combined with high-specificity Cas9 variants, and require minimal expenditure of time, effort, and resources. To this end, we developed an orthogonal and general approach for suppressing off-targets that can be readily combined with existing methods, including high-specificity variants.” (Page 2, Paragraph 2)

- “For example, dOTS could be used to address refractory off-targets of the popular engineered high-specificity Cas9 variants^{18–22,51,52}. Here, we showed that dOTS could effectively suppress editing at four refractory off-target sites with three high-specificity Cas9 variants.” (Page 7, Paragraph 2)
- Genome-wide off-target information is not necessary for all applications. dOTS can be used to suppress a problematic off-target that may interfere with specific research applications (e.g. editing of CCR2 by a sgRNA designed to target CCR5, or editing of HBD by a sgRNA designed to target HBB) that was identified computationally or empirically.

As discussed in our response to critique 1 above, dOTS is able to suppress a wide range of off-targets.

Suppression of multiple off-targets is possible as has been demonstrated in the manuscript. In those experiments, no dRNA titrations were needed. The limitations of multiplexing are discussed in the text: “Another drawback is that some dRNAs decrease on-target editing, particularly when they are multiplexed to suppress several off-target sites simultaneously. We suspect that these losses in on-target editing likely arise due to dilution of the plasmids or competition between sgRNAs and dRNAs to complex with Cas9. The first issue could be addressed by using a multiplex guide expression scheme^{46,47}, and both could be addressed by delivering preformed ribonucleoprotein mixtures⁴⁸” (page 7, paragraph 1). Thus, we feel that these limitations have been fully and fairly discussed.

3. Is there any rule to choose the candidate dRNAs? Did the authors test all potential dRNAs as long as PAM sequences are available? How does the distance between the off-target cleavage site and dRNA binding site affect the performance of dRNAs? How did the authors pick the length of dRNA spacers among 14 to 16 nt? It would be good if the authors give some guidelines for designing the best dRNAs.

- For the majority of sites included in this study (15/19 sites), we were able to identify effective dRNAs for suppressing off-target editing. Therefore, we did not perform exhaustive screening of all candidate dRNAs at most sites. We agree with the reviewer that the manuscript would benefit from providing guidelines for designing dRNAs. We have added a section in the methods to describe the design of dRNAs:

“dRNA sequences were designed by identifying 14 to 16 nucleotide dRNA spacer sequences which met the following criteria: (1) the dRNA spacer sequence and/or its PAM overlaps with the off-target spacer sequence and/or its PAM, and (2) the dRNA spacer or PAM exhibits perfect complementarity to the off-target but not the on-target locus. Spacer sequences with a 5' G were preferentially selected, but spacers containing a mismatched 5' G were also

used. Exhaustive screening of all candidate dRNAs which met the criteria was not performed at all sites. Alignments of the on-target sequences, off-target sequences, and dRNAs used in this study are presented in **Supplementary Figure 4**” (page 8, paragraph 4).

At present, there are no clear rules for predicting effectiveness of dRNAs and they must be identified empirically. However, at most of the sites we tested, only a few dRNAs needed to be screened to identify an effective dRNA for most off-targets. Discerning the design rules for effective off-target suppression is an exciting area for future study.

4. Fig 2. Why dRNA RNP 1350 fmoles shows higher OT cleavage than lower concentrations? The unit is missing on the right plots for both a and b panels.

- While *in vitro* studies are useful for studying certain mechanistic aspects of Cas9 activity, they do not fully capture the activity of Cas9 in the cellular environment. We do not know why there is a slight increase in off-target editing at 1350 dRNA but we observe that an excess of dRNA to sgRNA does not lead to a similar increase in off-target editing in cells (see Figs 3b and S8). Indeed, we observe that there is a slight decrease in off-target editing, as would be expected. Therefore, this result appears to be specific to nuance in the *in vitro* assay.

5. Fig 4. Y-axes need to be properly labeled. The reduction in the on-target activity in the case of eSpCas9 needs to be explained by considering the fact that lower DNA binding affinity of eSPCas9 can alter competition between sgRNA and dRNA RNPs.

- We are not aware of an issue with the y-axes in these plots, as they are consistent with plots displayed throughout our manuscript. Perhaps there was an issue with the PDF on the reviewer’s computer? The axis labels in the original submission were “Indel frequency (%)” on the top panels and “On-target/Off-target indel frequency ratio” on the bottom panels, and we think these are correct.

For the FANCF sgRNA2 locus, we do see a decrease for eSpCas9 with addition of a dRNA. However, this is not a uniform phenomenon as we do not see decreases in on-target efficiency for all loci (Supplementary Fig. 10). We address the decrease in editing at some targets in the text: “Indeed, in some cases, we observe a decrease in on-target editing when high-specificity Cas9 variants and dOTS are combined. However, this reduction in on target editing is generally less pronounced than the efficiency loss observed comparing HypaCas9 or SpCas9-HF1 to wild-type in the absence of dOTS” (page 5, paragraph 2).

6. Fig 5. Error bar in the panel b is beyond acceptable value, needs to be replaced.

- There are no error bars in Fig. 5, so we are unsure to what the reviewer is referring.

We do note that there is high variance in a minority of experiments, including in Fig. 5b. However, the high variance seen in this particular experiment does not impede our ability to make the basic conclusion that multiplex dOTS is effective. This is clearly seen in the rightmost plot in Fig. 5b, where the data clearly shows the pronounced specificity improvement for VEGFA sgRNA2 at both OT1 and OT17. Therefore, we decline to repeat this experiment.

7. SI, Fig 2, panel d: results indicate that the introduction of dRNA1 results in a complex situation. Apart from reducing OT1 activity and enhancing on-target activity it also increased off-target activity at other sites. Authors need to discuss the observation and their implication on the dRNA approach in a broader perspective.

Reviewer 2 notes that we observe an increase in the number of GUIDE-seq read counts at some sites in the presence of dRNA1 for the GUIDE-seq experiment presented in Figure S2. We agree that this is an interesting result but it is not clear what sort of conclusions can be drawn from this observation. Our interpretation of the guide-seq experiment is that we obtain greater editing in general (both for the on-target site and for off-target sites that are not shielded by dRNA1) when dRNA1 is included with our transformation. However, this does not appear to be a general result because for a vast majority of our other experiments the inclusion of dRNAs does not affect overall editing efficiency. Rather, this could be the result of higher electroporation efficiency of Cas9 RNPs and/or the dsODN tags in the sgRNA plus dRNA1 condition. This interpretation is more consistent with results elsewhere in the manuscript. In our multiplex dOTS experiments, we found that editing at each off-target site was only impacted by its cognate dRNA, not by dRNAs which target other off-targets for the sgRNA being used (Supplementary Fig. 11). In other words, the dRNAs were very selective for their off-targets and had no detectable impact on the other sites. We now describe this in the text: “Notably, each dRNA only impacted editing at its cognate off-target site, without increasing or decreasing the editing at the other off-target sites of the sgRNA” (page 2, paragraph 3).

Consistent with our interpretation, GUIDE-seq reads do not correlate well with indel frequency and can identify false positives, particularly at low GUIDE-seq read counts (PMID: 31000663). Indeed, there is far less agreement between genome-wide off-target detection methods, e.g. BLISS, Digenome-seq, and GUIDE-seq, at these weaker, lower confidence putative off-target sites (PMID: 28497783).

Due to the greater overall Cas9 activity and/or dsODN tag integration observed in the presence of dRNA1, the data presented in Supplementary Figure 11, and the limitations of GUIDE-seq and other genome-wide off-target profiling methods, we cannot conclusively rule in or rule out that editing is actually increasing at these weak off-targets relative to the sgRNA.

We can conclude, however, that dRNA1 is not inducing cleavage anywhere in the genome on its own. For the reasons above, we have not modified the text as Reviewer 2 has requested. We feel that we can't really say much beyond what is already stated.

8. Blocking the recutting of the HDR-repaired gene by dRNA is a good strategy, however, its application is limited by the narrow scope of the approach itself. The authors need to present a diagram explaining the gene target, HDR-repaired (BFP to GFP) gene sequence and dRNA alignment with the repaired product for a better understanding. While the author reports mutation from CAT to TAT codon for BFP to GFP editing, however, the mentioned reference (Nature Biotechnology, 2016, 34, 339–344) suggests changing to TAC codon instead. In addition, reference work also suggests mutation in the PAM sequence to impair recutting. The authors need to present the design clearly.

- We now include an alignment of the dRNA to the native and repaired sequence in Figure 6 to more clearly communicate the experimental design. We believe the text is clear in describing that previous approaches, including in the cited reference, relied on making multiple blocking mutations. In this reference, the T to C mutation in the 3rd position of the codon is a blocking mutation whose only purpose is to prevent recutting by Cas9. dReCS enables conversion of BFP to GFP **without** the blocking mutations that were needed in previous reports.

“Previously, several blocking mutations were used to prevent recutting, yet only a single nucleotide change is needed to alter the His in BFP (CAT) to the Tyr in GFP (TAT)⁴³.” (Page 6, paragraph 2)

9. Supplementary figures 6 and 7 didn't appear on the manuscript.

- We believe the reviewer was mistaken, as Supplementary Figures 6 and 7 (now Supplementary Figures 7 and 8) are both called out in the manuscript:
 - “Finally, we found that dRNA-mediated suppression of off-target editing was durable, with dRNAs effectively decreasing off-target editing for at least 72 hours post-transfection (**Supplementary Fig. 7**)” (page 3, paragraph 2).
 - “Consistent with our self-competition mechanism, preincubation of the substrate with the Cas9•sgRNA2 complex followed by addition of the

Cas9•dRNA1 complex eliminated the reduction in cleavage (**Supplementary Fig. 8**). Thus, Cas9•dRNA complexes can directly shield off-target loci from Cas9•sgRNA cleavage” (page 4, paragraph 1).

10. It seems like that fine-tuning the expression level of gRNA and dRNA is important for maximizing the on-target editing and minimizing off-target indels. This would be easier when Cas9 is delivered in the form of a ribonucleoprotein (RNP) complex. Moreover, RNP delivery is a better option for reducing off-target effects. In fact, the authors already described that RNP delivery could be an alternative for keeping the on-target activity in multiplexing and for the allele-specific editing. So, it would be great if the authors can present data from RNP delivery-based dOTS method, but it's not a must.

- As shown in the manuscript, fine-tuning of expression was not necessary for identifying an effective dRNA at any of the 15 sites identified in the paper, but could potentially be used to improve performance if warranted. We agree that using dOTS and dReCS with RNP delivery is an exciting application, however it is outside the scope of this initial study. We discuss the potential of using RNP delivery for dOTS in two sections of the manuscript:
 - “Based on the principle of Cas9 self-competition, electroporation of Cas9•dRNA RNPs to quench editing by the active Cas9•sgRNA RNP should allow fine tuning of editing efficiencies” (page 7, paragraph 3).
 - “Another drawback is that some dRNAs decrease on-target editing, particularly when they are multiplexed to suppress several off-target sites simultaneously. We suspect that these losses in on-target editing likely arise due to dilution of the plasmids or competition between sgRNAs and dRNAs to complex with Cas9. The first issue could be addressed by using a multiplex guide expression scheme^{46,47}, and both could be addressed by delivering preformed ribonucleoprotein (RNP) mixtures⁴⁸” (page 7, paragraph 1).